# Widespread Changes in the Immunoreactivity of Bioactive Peptide T14 After Manipulating the Activity of Cortical Projection Neurons

**DOI:** 10.3390/ijms26125786

**Published:** 2025-06-17

**Authors:** Auguste Vadisiute, Sara Garcia-Rates, Clive W. Coen, Susan Adele Greenfield, Zoltán Molnár

**Affiliations:** 1Department of Physiology, Anatomy and Genetics, Sherrington Building, University of Oxford, Parks Road, Oxford OX1 3PT, UK; auguste.vadisiute@dpag.ox.ac.uk; 2St John’s College, University of Oxford, St Giles’, Oxford OX1 3JP, UK; 3Neuro-Bio Ltd., Building F5, Culham Campus, Abingdon OX14 3DB, UK; sara.garciarates@neuro-bio.com; 4Faculty of Life Sciences and Medicine, King’s College London, London SE1 1UL, UK; clive.coen@kcl.ac.uk

**Keywords:** 14-mer peptide (T14), Synaptosomal-associated protein 25 kDa (Snap25), Retinol binding protein 4 (Rbp4), designer receptors exclusively activated by designer drugs (DREADDs), developing brain, Alzheimer’s disease, immunoreactivity

## Abstract

Previous studies have suggested that T14, a 14-amino-acid peptide derived from acetylcholinesterase (AChE), functions as an activity-dependent signalling molecule with key roles in brain development, and its dysregulation has been linked to neurodegeneration in Alzheimer’s disease. In this study, we examined the distribution of T14 under normal developmental conditions in the mouse forebrain, motor cortex (M1), striatum (STR), and substantia nigra (SN). T14 immunoreactivity declined from E16 to E17 and further decreased by P0, then peaked at P7 during early postnatal development before declining again by adulthood at P70. Lower T14 immunoreactivity in samples processed without Triton indicated that T14 is primarily localised intracellularly. To explore the relationship between T14 expression and neuronal activity, we used mouse models with chronic silencing (Rbp4Cre-Snap25), acute silencing (Rbp4Cre-hM4Di), and acute activation (Rbp4Cre-hM3D1). Chronic silencing altered the location and size of intracellular T14-immunoreactive particles in adult brains, while acute silencing had no observable effect. In contrast, acute activation increased T14+ density in the STR, modified T14 puncta size near Rbp4Cre cell bodies in M1 layer 5 and their projections to the STR, and enhanced co-localisation of T14 with presynaptic terminals in the SN.

## 1. Introduction

T14 is a peptide derived from the widespread enzyme acetylcholinesterase (AChE) but independently bioactive of the parent molecule and cholinergic transmission. T14 promotes cell growth via the mTOR pathway and declines with normal ageing. It acts exclusively at an allosteric site on alpha-7 nicotinic receptors to increase Ca²⁺ entry in the presence of the primary ligand acetylcholine (ACh) [1,2]. Its bioactivity has been demonstrated *in vitro* and *ex vivo* using PC12 cells and rodent brain slices. It has been hypothesised that T14 acts as an activity-dependent signalling molecule, promoting cell growth and renewal during brain development, but when inappropriately activated in the mature brain, it may instead be the key molecule driving the process of neurodegeneration in diseases such as Alzheimer’s disease (AD) [2,3]. In this context, previous *in vivo* studies in the 5XFAD mouse model of AD have shown elevated T14 immunoreactivity in the rodents’ brains in the areas implicated in neurodegenerative diseases [4,5].

To further investigate T14 under normal developmental conditions and in the adult brain, we explored its immunoreactivity in the mouse forebrain at embryonic days 16 and 17 (E16, E17), as well as postnatal days 0 (P0), 7 (P7), and 70 (P70). We examined T14 expression throughout the forebrain, with a focus on the primary motor cortex (M1), striatum (STR), and substantia nigra (SN).

To better understand the cellular localization of T14, we employed immunohistochemical labelling with and without Triton, a detergent that permeabilizes cell membranes and thereby reveals intracellular distribution. Previous studies have suggested that T14 expression and release may be activity-dependent [2]. Therefore, we tested the hypothesis that T14 levels and distribution are regulated by neuronal activity using *in vivo* mouse models involving both chronic and acute manipulation of cortical projection neuron activity.

For chronic silencing, we used the Snap25 conditional knockout (cKO) transgenic mouse model, in which regulated synaptic vesicle release is abolished in cortical layer 5 Rbp4^Cre^-expressing neurons and their projection targets [6,7,8]. Synaptic vesicle release is controlled by the SNARE complex, which plays a crucial role in membrane fusion, neurotransmitter release, intracellular trafficking, and neuronal maintenance [9,10,11].

Snap25, a central component of the SNARE complex, acts as an anchor and is essential for regulated synaptic vesicle release [12]. It is also implicated in cognitive function, long-term memory consolidation, and neuronal survival [13,14,15,16,17]. In our Snap25 cKO model, Snap25 expression is abolished in Rbp4 cortical layer 5 projection neurons via Cre-recombinase-mediated excision of exons 5a and 5b [8]. The model has been validated electrophysiologically [8] and the effects of chronic silencing on synapses, myelination, cell death, and behaviour have been documented [6,7,8,18].

Additionally, we employed acute manipulation of neuronal activity using designer receptors exclusively activated by designer drugs (DREADDs). These were targeted to the same Rbp4^Cre^-expressing cortical layer 5 neurons. Using clozapine-N-oxide (CNO), we acutely inhibited or activated neuronal activity for durations ranging from minutes to hours [18]. Activation of inhibitory DREADD (hM4Di) by CNO induces hyperpolarization and suppresses synaptic vesicle release, while activation of excitatory DREADD (hM3Dq) increases neuronal excitability via protein kinase C activation and intracellular Ca²⁺ signalling [19,20]. Both hM4Di and hM3Dq were genetically expressed in a Cre-dependent manner [18].

In this study, we investigated how chronic silencing of a subpopulation of cortical layer 5 neurons affects T14 distribution in M1 layer 5, STR, and SN, and how T14 co-localises with presynaptic terminals across different brain regions.

## 2. Results

### 2.1. Intracellular Immunoreactivity of T14 Peptide in Developing and Adult Brains 

To examine the distribution of T14 during normal development and in adult mouse brains, we used C57BL/6 mice and a previously characterised anti-T14 antibody [4,21,22], which was raised against the 14-mer peptide derived from the C-terminus of AChE. T14 immunoreactivity was quantified as the mean immunofluorescent intensity across entire coronal brain sections at embryonic days 16 and 17 (E16, E17) and postnatal days 0 (P0), 7 (P7), and 70 (P70) (Figure 1a). To investigate the cellular localisation of T14, we performed immunohistochemistry on free-floating coronal brain sections under two different conditions to distinguish between intracellular and extracellular localization. Sections were incubated in blocking solutions containing either 10% normal goat serum (NGS) alone or 10% NGS with 0.3% Triton X-100, a detergent commonly used to permeabilize cell membranes. The presence of Triton (+Triton) disrupts the integrity of the plasma membrane, allowing antibodies to access intracellular compartments. This condition enables the detection of both intracellular and extracellular antigens. In contrast, the absence of Triton (-Triton) preserves the membrane barrier, limiting antibody access to extracellular or membrane-associated antigens only. By comparing T14 immunoreactivity between these two conditions, we can infer its subcellular localisation. Results showed a statistically significant lower level of T14 immunoreactivity in the -Triton condition compared with +Triton across all developmental stages studied (E16, E17, P0, P7, P70) (F(1,8) = 3738, *p* < 0.0001; Figure 1c). Following Triton treatment, we observed that T14 immunoreactivity changes significantly during early development (F(4,10) = 532.9, *p* < 0.0001; Figure 1b). Specifically, immunoreactivity decreased at E17 compared with E16 and declined further by P0. During early postnatal development, T14 immunoreactivity peaked at P7 compared with P0, E16, and E17, and then decreased again by adulthood at P70, relative to both P7 and the embryonic stages. We further assessed T14 immunoreactivity at E16 and E17 in anterior and posterior regions of the embryonic brain. Significantly less T14 signal was observed in -Triton sections at E16 (F(1,8) = 13,024, *p* < 0.0001; Figure 1e) and E17 (F(1,8) = 3446, *p* < 0.0001; Figure 1f). In the +Triton sections, T14 immunoreactivity was significantly higher in the anterior cortex compared with the posterior cortex at E17 (F(1,8) = 25.46, *p* = 0.0010; Figure 1f). In postnatal brains, we examined T14 immunoreactivity in the M1, STR, and SN at P0, P7, and P70 (Figure 1g–k). At P0, significantly less T14 immunoreactivity was detected in all regions in the -Triton condition (F(1,12) = 939.7, *p* < 0.0001; Figure 1i); Triton treatment revealed a higher T14 signal in M1 compared with SN (F(2,12) = 28.53, *p* < 0.0001; Figure 1i). At P7, T14 immunoreactivity was significantly lower in all regions in the -Triton condition compared to +Triton (F(1,12) = 3653, *p* < 0.0001; Figure 1j); following Triton, there were no significant differences in T14 levels between the regions studied (F(2,12) = 0.7176, *p* = 0.5077; Figure 1j). In adult brains at P70, T14 immunoreactivity was reduced in all regions under the -Triton condition (F(1,12) = 463.0, *p* < 0.0001; Figure 1k), as observed at earlier stages; in the +Triton sections, it was significantly higher in the SN compared with M1 (F(2,12) = 5.110, *p* = 0.0248; Figure 1k). Together, these data show that T14 immunoreactivity is consistently lower in the absence of Triton across all developmental stages and brain regions examined, supporting the conclusion that T14 is primarily localised intracellularly. Furthermore, T14 expression peaks at P7 and undergoes region-specific changes during postnatal development, suggesting a dynamic, developmentally regulated pattern of intracellular localization in both developing and adult brains.

### 2.2. Abolishing Regulated Release of Synaptic Vesicles from a Subpopulation of Layer 5 Projection Neurons Changes the Distribution of T14 Immunoreactivity (488)

To better understand whether T14 peptide expression is activity-dependent, we used a Snap25 conditional knockout (cKO) mouse model. In this transgenic model (Figure 2b), regulated synaptic vesicle release was abolished in a subpopulation of cortical layer 5 Rbp4Cre-expressing neurons and their projection regions. Rbp4Cre is active from early development through to adulthood [8]. T14 peptide distribution was investigated in this model using immunohistochemistry [23,24]. We studied adult mice at 8 months of age and measured the density and size of T14 immunoreactivity (Figure 2a), as well as its co-localisation with presynaptic terminals, identified by vesicular glutamate transporters 1 and 2 (vGlut1 and vGlut2, Figure 2g), in M1 layer 5 (L5), the striatum (STR), and the substantia nigra (SN)—the latter lacking direct projections from cortical layer 5 neurons. T14 density in M1 L5 was unchanged in Snap25 cKO mice compared with controls (*t* = 1.044, *df* = 4, *p* = 0.3556) but was significantly decreased in the STR (*t* = 18.76, *df* = 4, *p* < 0.0001; Figure 2c) and increased in the SN (*t* = 5.046, *df* = 4, *p* = 0.0073; Figure 2c). We observed that T14 formed immunoreactive clusters in Snap25 cKO mice compared to controls.

To better understand these changes, we quantified T14 immunoreative puncta size and categorised them as small (<0.5 μm^2^), medium (0.5–<1 μm^2^), or large (≥1 μm^2^). In M1 L5, the average size of T14+ puncta did not differ significantly between groups, but there was a reduction in the proportion of medium-sized puncta in Snap25 cKO mice compared with controls (F(2,12) = 534, *p* < 0.0001; Figure 2d). In the STR, the average size of T14+ puncta was increased in Snap25 cKO mice (*t* = 7.951, *df* = 4, *p* = 0.0014), accompanied by a decrease in the proportion of small puncta and an increase in large puncta (F(2,12) = 2694, *p* < 0.0001; Figure 2e). No changes in T14+ puncta size were observed in the SN (Figure 2f). Given that regulated synaptic vesicle release is abolished in this transgenic model, we next examined T14 co-localisation with presynaptic markers (Figure 2g). No differences in T14 co-localisation with vGlut1 were observed in Snap25 cKO mice compared with controls (Figure 2h,i). However, in the SN, T14+ puncta co-localisation with vGlut2 was significantly reduced in Snap25 cKO mice (*t* = 3.435, *df* = 4, *p* = 0.0264; Figure 2j). These findings suggest that chronic silencing leads to the formation of T14 immunoreactive ‘clusters’ that increase in size in silenced M1 L5 neurons and their projections to the STR, while affecting co-localisation with presynaptic terminals specifically in the SN.

### 2.3. Acute Chemogenetic Manipulation Affects the Distribution of T14 Immunoreactivity and Presynaptic Compartments (430)

To further examine the role of T14 in an activity-dependent manner, we used acute chemogenetic manipulation of the same layer 5 neuronal population (Rbp4Cre). Specifically, we established adult inhibitory and excitatory DREADD mouse models [18]. DREADD-based methods allow for the chemogenetic manipulation of activity in well-defined neuronal populations. In this model, we genetically expressed hM4Di (inhibitory DREADD) and hM3Dq (excitatory DREADD) in cortical neurons using the Rbp4Cre mouse line, as previously described [18]. All adult mice were perfused 90 min after intraperitoneal (IP) injection of either saline or clozapine-N-oxide (CNO), based on our previous findings that CNO effects peak approximately 1 h post-injection [18]. Rbp4Cre-hM4Di (inhibitory DREADD) mice were injected with either saline or 5 mg/kg CNO, and Rbp4Cre-hM3Dq (excitatory DREADD) mice received saline or 10 mg/kg CNO [18]. We measured T14+ puncta density and size, as well as T14+ puncta co-localisation with presynaptic compartments immunolabelled with vGlut1 and vGlut2 in M1 L5, STR, and SN in both inhibitory (Rbp4Cre-hM4Di, Figure 3a) and excitatory (Rbp4Cre-hM3Dq, Figure 4a) DREADD models. In the inhibitory DREADD model, no statistically significant changes were observed in T14 immunoreactivity, puncta size, or co-localisation with presynaptic terminals (Figure 3a–h), suggesting that acute silencing may not impact T14 distribution within soma or synaptic terminals. However, changes were observed following acute activation of cortical neurons. In the Rbp4Cre-hM3Dq model, T14 density was significantly increased in the STR following 10 mg/kg CNO treatment (*t* = 4.889, *df* = 4, *p* = 0.0081; Figure 4b,c). Although the average T14 puncta size was not significantly altered in M1 L5, STR, or SN after CNO treatment (Figure 4e), we observed a shift in the size distribution of T14+ puncta. In M1 L5, the proportion of small puncta increased, while medium and large puncta decreased (F(2,12) = 4779, *p* < 0.0001; Figure 4f). Conversely, in the SN, the proportion of small puncta decreased, while medium and large puncta increased following CNO administration (F(2,12) = 3894, *p* < 0.0001; Figure 4f). We next assessed changes in T14+ co-localisation with vGlut1 and vGlut2 immunoreactivities (Figure 4g). No significant differences were observed in T14+ co-localisation with vGlut1 in M1 L5 or STR. However, in the SN, T14+ co-localisation with vGlut2 was significantly increased following CNO treatment (*t* = 4.031, *df* = 4, *p* = 0.0157; Figure 4h).

In summary, activation of the inhibitory DREADD had no effect on T14 distribution. In contrast, activation of the excitatory DREADD affected T14+ density in the STR (a projection target of Rbp4Cre neurons), altered T14 puncta size near Rbp4Cre cell bodies in M1 L5 and in projections to the STR, and increased T14 co-localisation with presynaptic terminals in the SN—a region that does not receive direct projections from cortical layer 5 neurons.

## 3. Discussion

In this study, we examined the immunoreactivity of a bioactive peptide that exhibits dual trophic toxicity, depending on the context—whether it is expressed during development or pathologically in maturity [2]. We explored changes during embryonic and postnatal brain development and demonstrated how chronic and acute manipulations of the activity of selected cortical layer 5 neuronal populations influence T14 immunoreactivity and morphology. Using C57BL/6 mice, we found that T14 immunoreactivity was most prominent in the P7 brains, differing from embryonic (E16/17), early postnatal (P0), and adult (P70) stages. We further used chronic and acute manipulations of Rbp4-Cre neuronal activity and observed changes in T14 immunoreactivity patterns in both models. Chronic silencing affected the density of T14 immunoreactivity in the STR and SN, T14+ puncta size in M1 and STR, and T14+ puncta co-localisation with vGlut2 in the SN. In contrast, acute silencing for 90 min had no effect on T14 immunoreactivity or puncta size, but acute activation influenced T14 immunoreactivity in the STR, T14+ puncta size in M1, and T14+ puncta co-localisation with vGlut2 in the SN.

### 3.1. T14 Immunoreactivity

The subcellular localization of T14 has not been fully resolved. Our results suggest that T14 is more likely to be intracellular than extracellular during embryonic and postnatal development and in the adult brain. This supports the idea that T14 may be released from neurons in an activity-dependent manner. We also observed that T14 forms granules. However, the location of the immunoreactivity associated with these granules in the cytoplasm has yet to be identified. These granules could be in the ER, Golgi, endosomes, lysosomes, or even aggregates that contain various proteins, with T14 being just one component. Ultrastructural analysis is needed to address this question.

In a previous study, we demonstrated the ultrastructure of degenerating neurons in the primary somatosensory cortex of 8-month-old Snap25 cKO brains. Using transmission electron microscopy (TEM) of Rbp4-Cre;Snap25fl/fl cortex that was immune-gold labelled for tdTomato, we observed tdTomato-positive neurons with altered cytoplasmic appearances. These tdTomato+ neurons contained dense cytoplasm packed with multivesicular bodies and vacuoles containing electron-dense material, which are putative lysosome-like structures—a characteristic feature of dystrophic neurons in neurodegenerative disorders such as Alzheimer’s disease [25,26,27,28]. The morphology of vacuoles and the electron density of the cytoplasm varied among neurons, possibly reflecting degeneration. It is possible that the observed vacuoles contain T14 immunoreactive peptides in these aggregates. However, we did not observe such aggregates following acute chemogenetic manipulations (stimulation or inhibition).

### 3.2. T14 Immunoreactivity After Chronic and Acute Manipulations of Neuronal Activity

To test whether neuronal activity plays a role in the distribution of T14, we used several mouse models where activity was modified in a chronic or acute manner [8,18]. In the Rbp4-Cre:Snap25fl/fl mice, a subset of cortical layer 5 neurons and dentate gyrus neurons lose their ability to release synaptic vesicles in a regulated fashion from late embryonic stages [8]. This affects both the activity of the cells and the activity elicited by their long-range terminals, influencing synapse development [24], synapse maintenance [8], myelination [6], and cell survival [8]. The sequence of synapse loss, myelination deficits, and cell death is dependent on the specific neuronal populations that are silenced, with layer 5 neurons being the most sensitive in the lines studied (Ntsr1-Cre and Drd1a-Cre) [7,8].

In exploring the well-characterised Snap25 cKO (Rbp4-Cre) model, we show that chronic silencing leads to changes in the location and size of T14 immunoreactive intracellular particles. These changes appear to be region-specific, with T14 immunoreactive particles only increasing in size in silenced M1 L5 and in L5 projections to the STR. Interestingly, T14 co-localisation with presynaptic terminals (vGlut2) only changed in the SN, which receives no direct Rbp4-Cre-positive projections. The Rbp4-Cre:Snap25*^fl/fl^* neurons exhibit slow degeneration, with neuronal loss becoming apparent from 8 months of age [8]. While degenerating and dying layer 5 neurons have been studied using EM [8,24], the exact mechanisms of their death remain unclear.

Using Rbp4Cre-hM3Dq and Rbp4Cre-hM4Di mice, we demonstrate that acute silencing (90 min after CNO application) does not alter T14 immunoreactivity patterns, while acute activation (within the same timescale) leads to an increase in T14+ density in the Rbp4-Cre projection region (STR) and changes in the size of T14 immunoreactive granules near Rbp4-Cre cell bodies in M1 L5 and projections to the STR. Furthermore, acute activation affects T14 co-localisation with presynaptic terminals (vGlut2) in the SN, a region that does not receive direct projections from cortical layer 5 neurons.

It appears that chronic silencing exerts a stronger effect on T14 than acute manipulations of neuronal activity. This suggests that prolonged disruption of synaptic function leads to more pronounced alterations in T14 expression or localization. In the absence of Snap25, other SNARE proteins or synaptic components may partially compensate for the loss; however, this compensation is often incomplete and insufficient to fully restore normal synaptic function, which may contribute to the progressive neuronal dysfunction observed [12,29,30].

## 4. Materials and Methods

### 4.1. Breeding and Maintenance

All experiments were approved by a local ethical review committee and conducted in accordance with the UK Animals (Scientific Procedures) Act, 1986 (ASPA), under valid personal and project licences. All animals were held in individually ventilated cages (IVCs) on a 12 h light/dark cycle in the Biomedical Sciences Building (BSB), Oxford University; water and food were given *ad libitum*. To evaluate intracellular and extracellular localisation of T14, C57BL/6 mouse brain at embryonic day (E)16, E17 and postnatal day 0 (P0), P7 and P70 was used. To understand activity-dependent changes of T14, we used three different transgenic mouse lines. Breeding to achieve conditional ablation of Snap25 expression from a subpopulation of cortex layer 5 neurons (Snap25 cKO) animals has been previously described [8]. For experiments comparing control and cKO animals, mice of the following genotypes were used: Cre/−;*Ai14*;Snap25*^fl/+^* (control) and Cre/+;Ai14;Snap25*^fl/fl^* (Snap25 cKO). The identification of Cre-positive mice was performed using fluorescence goggles (Biological Laboratory Equipment Maintenance and Service, BLS) and genotyping. Furthermore, we used a chemogenetic method and the layer 5 driver line Rbp4-Cre to develop acute activation mouse models as previously described [18]. The layer 5 driver mouse expressing Cre recombinase Tg(Rbp4-cre)KL100Gsat/Mmucd (Rbp4-Cre) was crossed with the inhibitory DREADD line B6.129-Gt(ROSA)26Sortm1(CAG-CHRM4*,-mCitrine)Ute/J (The Jackson Laboratory, No: 026219, Bar Harbor, ME, USA) to generate recombinant ‘inhibitory DREADD’ mice abbreviated as Rbp4^Cre^-hM4Di and with the excitatory DREADD line B6N;129-Tg(CAG-CHR3*, mCitrine)1Ute/J (The Jackson Laboratory, No: 026220, Bar Harbor, ME, US) to generate recombinant ‘excitatory DREADD’ mice abbreviated as Rbp4^Cre^-hM3Dq. All genotyping was performed by Transnetyx.

### 4.2. Tissue Collection

Mice were anaesthetised with 0.6 mL/kg Pentobarbital administered via IP. Anaesthesia was confirmed by a pedal reflex test. Animals were perfused with 0.1 M phosphate-buffered saline (PBS, pH 7.4) and 4% formaldehyde (PFA diluted in PBS, F8775; Sigma Aldrich, Gillingham, UK). For embryonic brain collection, embryos were decapitated under Schedule 1 and the brain was dissected immediately. Embryonic and postnatal brains were removed and post-fixed in 4% PFA overnight at 4 °C and were transferred to 1× PBS with 0.05% sodium azide (26628-22-8; Sigma Aldrich) the following day and stored at 4 °C until further usage. PFA-fixed brains were embedded in 5% agarose (Bioline, DM50-113D, London, UK) and cut with a vibrating microtome (VT1000S, Leica Systems, Wetzlar, Germany) into 50 μm thick coronal sections.

### 4.3. CNO

Clozapine-N-oxide dihydrochloride (Torcis, 6329) was used to prepare CNO solution. CNO solution was prepared under sterile conditions and passed through a Millipore filter. CNO was kept at −20 °C and defrosted on the day of use and diluted using sterile saline. Inhibitory DREADD mice were injected with 5 mg/kg CNO or saline. Excitatory DREADD mice were injected with 10 mg/kg CNO or saline. 

### 4.4. Immunohistochemistry and Imaging

Free-floating brain sections were blocked with 2–10% normal goat serum (Sigma Aldrich, Gillingham, UK) with or without 0.2–0.3% TritonX-100 (Sigma Aldrich, Gillingham, UK) in PBS (blocking solution, 2 h at RT) and incubated with one of the following primary antibody combinations: rabbit anti-T14 (1:500, T14 validation Rocha et al., 2023 [21]); rabbit anti-T14 (1:500) and guinea pig anti-vGlut1 (1:500, Merck Millipore AB5905, Gillingham, UK); rabbit anti-T14 (1:500) and guinea pig anti-vGlut2 (1:500, Merck Millipore AB2251). Primary antibody combinations underwent overnight, 20 or 48 h incubation at 4 °C. Sections were then washed in 0.1 M PBS before incubating with secondary antibodies (goat anti-rabbit A488 (1:500, ThermoFisher A11034, Oxford, UK); goat anti-guinea pig A633 (1:500, Molecular Probes A21105, Eugene, OR, USA)) in blocking solution at room temperature for 2 h. Immunolabelled sections were counterstained with DAPI (1:1000, Invitrogen D1306, Paisley, UK) mounted with FluorSave^TM^ (Merck Millipore 345789, Gillingham, UK) and imaged as described below. 

### 4.5. Imaging

#### 4.5.1. Laser-Scanning Confocal Microscopy

All imaging and analysis were carried out blind to the animal age, genotype, and condition (with or without triton). To evaluate T14 immunoreactivity in brain sections with and without triton in the primary motor cortex (M1) (*n* = 3 images/animal), striatum (STR) (*n* = 3 images/animal), and substantia nigra (SN) (*n* = 3 images/animal), immunolabeled sections were imaged with a laser-scanning confocal microscope (Zeiss LSM710, Oberkonchen, Germany) using 20×/0.6 air objective and 1x optical zoom at 0.42 μm pixel size, and frame size 1024 × 1024. To examine T14 immunoreactive puncta density, size and co-localisation with vGlut1 or vGlut2 in M1 layer 5, STR and SN (*n* = 3 images/animal) immunolabeled sections were imaged with a laser-scanning confocal microscope (Zeiss LSM710, Oberkonchen, Germany) using 63×/1.4NA oil-immersion objective and 2× optical zoom. 

#### 4.5.2. Image Processing and Analysis

T14+ immunofluorescence was analysed with ImageJ software 2.14.0 (NIH) and we manually adjusted threshold and measured mean grey value, and for T14 density, size, and vGlut1+ and vGlut2+ puncta number, we used 3D ROI manager after subtracting background to remove spatial variations in the background intensities for 3D puncta reconstruction, applying Gaussian blur 3D smoothing filter (x sigma = 2, y sigma = 2, z sigma = 2) and manual thresholding. Intensity measurements were normalised to control signal levels. Data was normalised based on number of planes in image and image volume. Co-localisation between T14 and vGlut1 or vGlut2 puncta was quantified using the ImageJ JACoP plugin and object-based method. All analysis and calculations are summarised in the reference guide provided for the JACoP plugin (https://imagej.net/plugins/jacop, accessed on 23 September 2024) and additional associated references [31].

### 4.6. Statistics

All analyses were performed on data from at least two brains per group. For each analysis, a minimum of three regions per brain, genotype, and CNO-injected condition were included for both inhibitory and excitatory DREADDs. The sample size (*n*) refers to individual mice for T14 mean intensity, density, size, and co-localisation with either vGlut1 or vGlut2. Data are reported as means ± standard error of the mean (SEM). Statistical analyses were performed using GraphPad Prism 10, with data normality verified using the Shapiro–Wilk test, assessed by QQ plot. For single comparisons between two groups in T14 co-localisation with vGlut1 or vGlut2, an unpaired Student’s *t*-test or Mann–Whitney test was used. Comparisons between three groups were conducted using one-way ANOVA with Tukey’s multiple comparison test, which was required for T14 immunofluorescent analysis in the developing and adult brains. Comparisons involving more than three groups were analysed using two-way ANOVA with Šidak’s multiple comparison test, which was required for analysis of T14 immunoreactivity in groups with and without Triton and T14+ immunoreactive puncta density, size, and co-localisation between T14 and vGlut1 or vGlut2 in saline and CNO-injected excitatory DREADD mice. For comparisons of more than three groups, a mixed-model ANOVA with Benjamini, Krieger, and Yekutieli multiple comparison test was used. This was required for analysis of T14+ immunoreactive puncta density, size, and co-localisation between T14 and vGlut1 or vGlut2 in control and Snap25 cKO mice, as well as in saline- and CNO-injected inhibitory DREADD mice. The threshold for statistical significance was set at *p* < 0.05. More detailed statistical information is provided in the Appendix A. All figures were created using BioRender.com accessed on 20 August 2024 to 13 June 2025.

## 5. Conclusions

We investigated the bioactive peptide T14, which has both trophic and toxic roles depending on developmental or pathological context. T14 immunoreactivity peaks at postnatal day 7 and varies across developmental stages. Chronic silencing of cortical layer 5 neurons in the Snap25 cKO model caused region-specific changes in T14 localization, density, and granule size, especially in the motor cortex and striatum, while acute silencing had little effect. Acute activation increased T14 levels and altered its association with presynaptic terminals.

Ultrastructural analysis revealed lysosome-like structures in degenerating neurites and cell bodies that may contain T14, linking chronic synaptic disruption to neuronal degeneration. These changes were not seen after acute manipulations. Partial compensation by other SNARE proteins occurs in the absence of Snap25 but is insufficient to prevent progressive dysfunction.

Our study underscores the complex role of T14 in neuronal development and pathology, highlighting the importance of synaptic activity patterns in modulating its function. Future work focusing on the precise subcellular localization of T14 granules and the mechanisms underlying neuronal death in Snap25-deficient neurons will provide deeper insight into its dual roles and therapeutic potential.

### Future Directions

T14 may serve as a promising therapeutic target and we have previously demonstrated that T14 levels vary across Braak stages, correlating with Alzheimer’s pathology progression [2]. However, it remains unclear whether T14 is expressed in cell bodies or terminals, and what triggers its release from cells. Electron microscopy may help resolve T14’s precise localization in the developing and adult brain. Since T14 selectively binds to alpha-7 receptors [1], we speculate that T14 might function as a signalling molecule; however, further studies are needed to confirm this hypothesis. Additionally, it is crucial to understand the relationship between T14 and specific forms of neurodegeneration. In our transgenic mouse model, which abolishes synaptic vesicle release in a subset of projection neurons in cortical layer 5, the first signs of neurodegeneration appear after postnatal day 21, and we observed T14 forming immunoreactive clusters at 8 months. We hypothesise that T14 cluster formation may be driven by chronic synaptic silencing in a regulated fashion or could be released with synaptic vesicles. We also observed that T14 co-localises with vGlut1 and vGlut2. Further analysis is required to confirm whether this co-localisation occurs inside or outside synaptic terminals and whether T14 co-localises with other proteins. Investigating T14 co-localisation with different postsynaptic terminal receptors will also be an important next step.

## Figures and Tables

**Figure 1 ijms-26-05786-f001:**
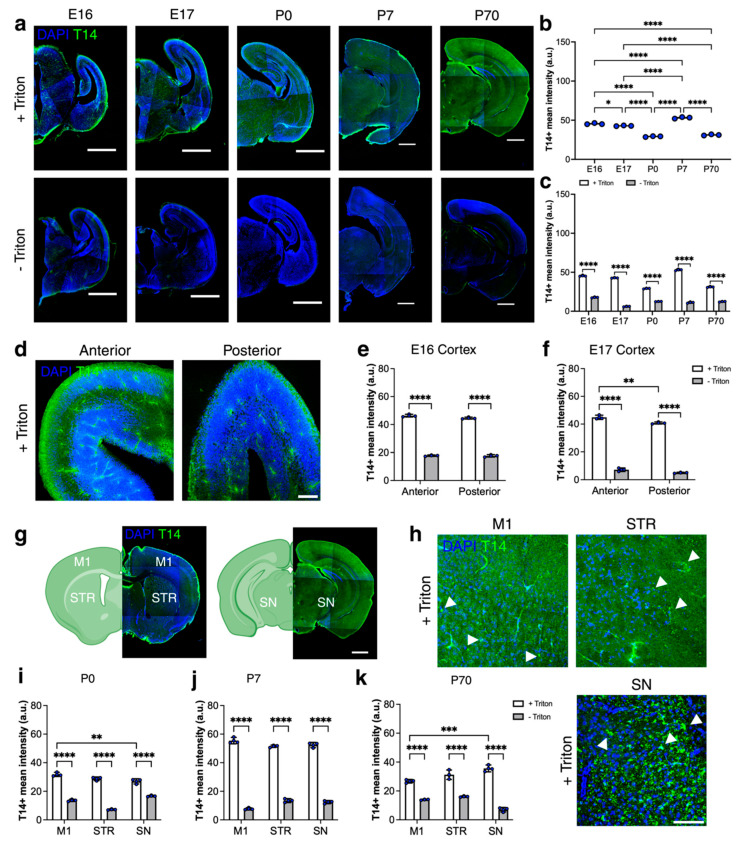
Changes in T14 immunoreactivity in the developing brains. (**a**) Representative images of T14 (green) immunostained brain sections at E16, E17, P0, P7, and P70, treated with or without Triton. (**b**,**c**) T14 immunoreactivity changes during development and in adult brains. *n* = 3 mice per age group; average of 3 sections per brain. Data evaluated using one-way ANOVA with Bonferroni’s multiple comparison test. T14 immunoreactivity with and without Triton in developing and adult brains was assessed using two-way ANOVA with Bonferroni’s multiple comparison test (*n* = 3 mice per age group; 3 sections per brain and condition). (**d**) Representative images of T14 (green) in the anterior and posterior regions of embryonic brains. (**e**,**f**) T14 immunoreactivity at E16 and E17 in anterior vs. posterior regions of sections treated with or without Triton. *n* = 3 mice per age group; average of 3 sections per brain and condition. Two-way ANOVA with Bonferroni’s multiple comparison test was used for analysis. (**g**,**h**) Representative images of T14 (green) in M1, STR, and SN, white arrows show T14 signal. (**i**–**k**) T14 immunoreactivity in M1, STR, and SN at P0, P7, and P70 in sections treated with or without Triton. *n* = 3 mice per age group; 3 sections per brain, per mouse, and per condition. Data were evaluated using two-way ANOVA with Bonferroni’s multiple comparison test. All data presented as mean ± SEM, * *p* < 0.05, ** *p* < 0.01, *** *p* < 0.001, and **** *p* < 0.0001 with scale bars of (**a**,**g**) 1000 μm and (**d**,**h**) 100 μm. Created with BioRender.com/egba8v9.

**Figure 2 ijms-26-05786-f002:**
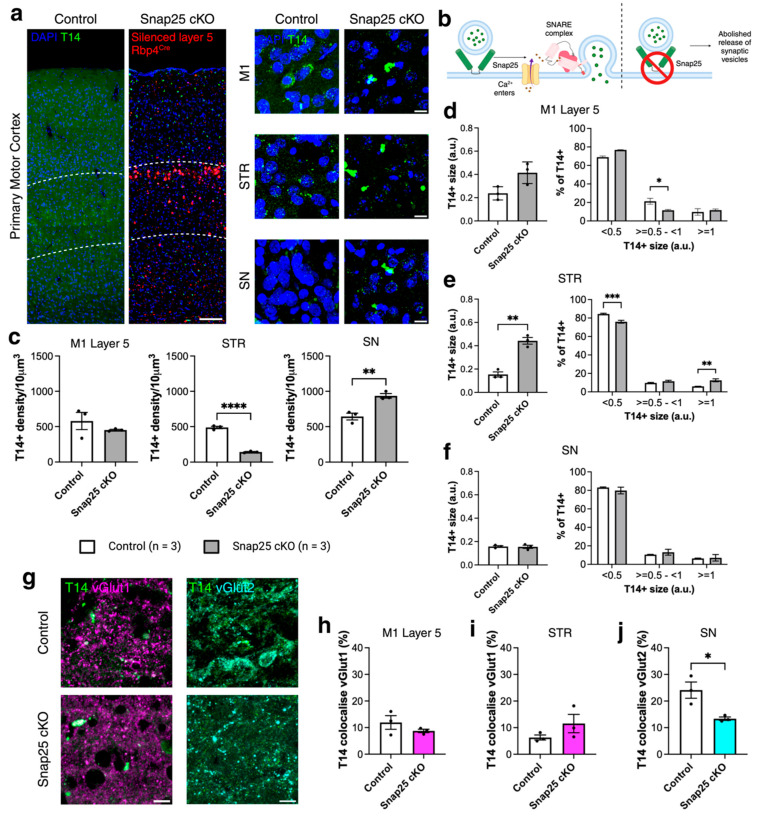
Chronic silencing of Rbp4^Cre^ subpopulation of layer 5 cortical projection neurons leads to T14-immunoreactive cluster formation in M1 L5, STR and SN. (**a**) Representative images of T14 (green) and silenced neurons (red) in M1 L5 (between white dash lines), STR, and SN at 8 months of age in control and Snap25 cKO mice. (**b**) Schematic of the Snap25 cKO model. (**c**) Changes in T14+ density in M1 L5, STR, and SN in control (white) and Snap25 cKO (grey) mice, with and without Triton. *n* = 3 mice per genotype; average of 3 sections per brain and per genotype. Data evaluated using *t*-tests. (**d**–**f**) Changes in T14+ puncta size in M1 layer 5, STR, and SN. *n* = 3 mice per genotype; average of 3 sections per brain and per genotype. Data evaluated using *t*-tests and two-way ANOVA with Bonferroni’s multiple comparison test. (**g**) Representative images of T14+ co-localisation with vGlut1 (magenta) or vGlut2 (cyan) in control and Snap25 cKO mouse brains. (**h**–**j**) T14+ puncta co-localisation with vGlut1 (magenta) in M1 layer 5 and STR, and with vGlut2 (cyan) in SN, in control and Snap25 cKO mice. *n* = 3 mice per genotype; average of 3 sections per brain and per genotype. Data evaluated using *t*-tests. All data presented as mean ± SEM, * *p* < 0.05, ** *p* < 0.01, *** *p* < 0.001, and **** *p* < 0.0001 with scale bars of (**a**) 1000 μm and (**g**) 100 μm. Detailed statistical information is listed in Appendix A. Created with BioRender.com/7yoqh7c.

**Figure 3 ijms-26-05786-f003:**
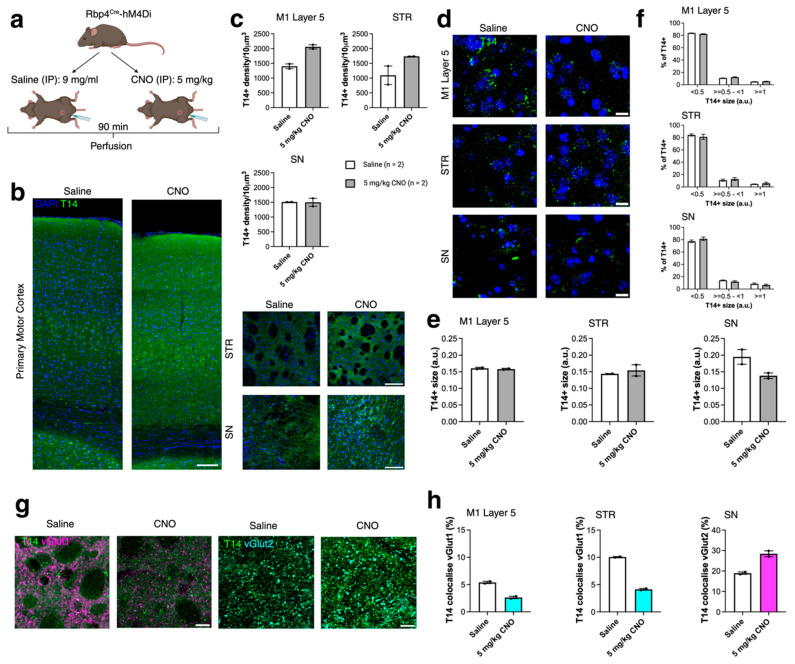
Acute chemogenetic silencing of Rbp4^Cre^ positive layer 5 projection neuron population has no effect on the distribution of T14 immunoreactivity. (**a**) Schematic of the inhibitory DREADD experimental procedure. (**b**) Representative images of T14 (green) in M1, STR, and SN of Rbp4Cre-hM4Di mice injected with saline or 5 mg/kg CNO. (**c**) T14+ puncta density in Rbp4Cre-hM4Di mice. *n* = 2 mice per condition; average of 3 sections per brain and per condition. Data evaluated using the Mann–Whitney test. (**d**) Representative images showing T14+ puncta size (green). (**e**,**f**) T14+ puncta size in M1 L5, STR, and SN after 5 mg/kg CNO treatment in Rbp4Cre-hM4Di mice. *n* = 2 mice per condition; average of 3 sections per brain and per condition. Data evaluated using the Mann–Whitney test and two-way ANOVA with Benjamini, Krieger, and Yekutieli multiple comparison correction, white color bar—saline, grey color bar—CNO injected. (**g**,**h**) T14+ puncta (green) co-localisation with vGlut1 (magenta) in M1 L5 and STR, and with vGlut2 (cyan) in SN in saline- and 5 mg/kg CNO-treated mice. *n* = 2 mice per genotype; average of 3 sections per brain and per genotype. Data evaluated using the Mann–Whitney test. All data presented as mean ± SEM, with scale bars of (**b**) 1000 μm and (**d**,**g**) 100 μm. Detailed statistical information is listed in Appendix A. Created with BioRender.com/uc2rpj9j.

**Figure 4 ijms-26-05786-f004:**
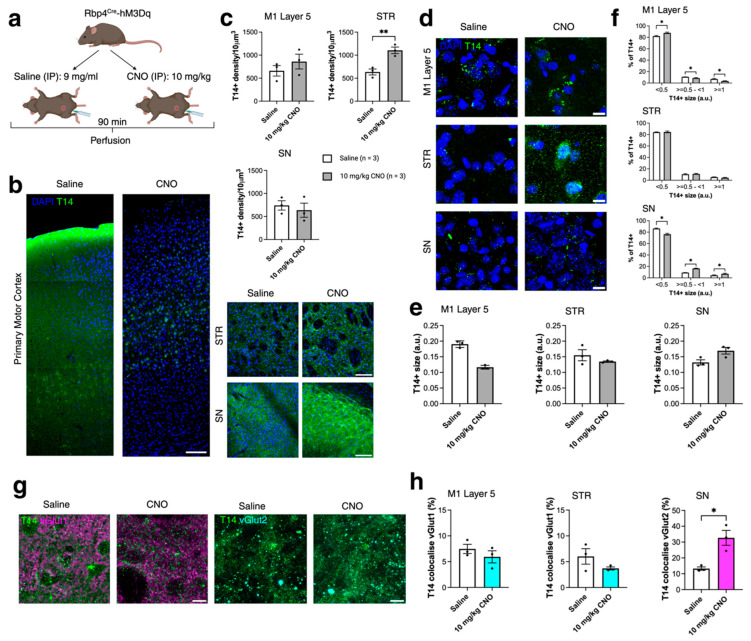
Acute chemogenetic activation of Rbp4^Cre^ subpopulation of layer 5 projection neurons affects the distribution of immunoreactivity for T14. (**a**) Schematics of excitatory DREADD experimental procedures. (**b**) Representative images of T14 (green) in M1, STR and SN of Rbp4^Cre^-hM3Dq mice injected with saline or 10mg/kg CNO. (**c**) T14+ density in Rbp4^Cre^-hM3Dq, *n* = 3 mice/condition, average of 3 sections per brain and per condition, evaluated using *t*-test. (**d**) Representative images of T14 puncta size (green). (**e**,**f**) T14+ puncta size after 10 mg/kg CNO treatment in M1 L5, STR and SN in Rbp4^Cre^-hM3Dq, *n* = 3 mice/condition, average of 3 sections per brain and per condition, evaluated using *t*-test and two-way ANOVA via Bonferroni’s multiple comparison test, white color bar—saline, grey color bar—CNO injected. (**g**,**h**) T14+ puncta (green) co-localisation with vGlut1 (magenta) in M1 L5 and STR, and with vGlut2 (cyan) in SN in saline and 10mg/kg CNO mice, *n* = 3 mice/genotype, average of 3 sections per brain and per genotype, evaluated using *t*-test. All data presented as mean ± SEM, * *p* < 0.05, ** *p* < 0.01, with scale bars of (**b**) 1000 μm and (**d**,**g**) 100 μm. Detailed statistical information is listed in Appendix A. Created with BioRender.com/oxnolcg.

## Data Availability

The original contributions presented in this study are included in the article/Appendix A. Further inquiries can be directed to the corresponding authors.

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
