# Peer review of "Widespread Changes in the Immunoreactivity of Bioactive Peptide T14 After Manipulating the Activity of Cortical Projection Neurons"

_ijms, 2025, doi:10.3390/ijms26125786_

Round 1
Reviewer 1 Report
Comments and Suggestions for Authors
Dear Authors,
Thank you for submitting your manuscript entitled "Widespread Changes of Bioactive Peptide T14 Immunoreactivity after Manipulation of Cortical Projection Neuron Activity". The study addresses an important and timely topic in the field of neurobiology and Alzheimer’s disease, particularly by exploring the activity-dependent regulation of T14, a peptide with emerging significance. The methodological design is conceptually sound and the results are of clear potential interest to the field.
However, several important issues must be addressed to improve the rigor, clarity, and translational relevance of the study. Below, I outline specific points that require your attention:
- Small sample sizes (n=2–3 mice per group). Increase the number of animals per group to at least 5–6 and perform a power analysis to justify sample sizes.
- Lack of biological replicates across batches. Repeat key experiments in at least two independent experimental batches or litters to ensure reproducibility.
- No validation of the anti-T14 antibody specificity. Perform pre-adsorption controls, Western blots using brain tissue, or test the antibody on T14 knockout samples.
- No biochemical confirmation of T14 distribution. Include Western blot or ELISA analyses to confirm the presence and size of T14 and support immunostaining results.
- Indirect assessment of intracellular vs. extracellular localization. Use immunogold electron microscopy or subcellular fractionation to directly visualize the location of T14.
- Co-localization analysis is largely descriptive. Quantify co-localization using Manders’ or Pearson’s coefficients and include appropriate statistical controls.
- No discussion of compensatory mechanisms in Snap25 Cko. Add a paragraph discussing potential compensatory changes due to early developmental silencing and suggest using inducible models in the future.
- No developmental time-course for T14 alterations. Include intermediate timepoints (e.g., P14, P30) to monitor the dynamics of T14 expression across development.
- No functional or behavioral readouts. Incorporate behavioral tests (e.g., open field, rotarod) or physiological recordings to connect molecular changes with brain function.
- No mechanistic explanation for changes in the substantia nigra. Discuss possible indirect circuit mechanisms (e.g., polysynaptic pathways or diffusible factors) and consider using retrograde tracing.
- No downstream pathway analysis. Investigate the impact of T14 on calcium signaling, α7 receptor activation, or downstream gene expression using pharmacological tools.
- Single-dose/single-timepoint chemogenetic manipulations. Explore a dose–response curve and test multiple timepoints (e.g., 30 min, 2h, 6h) to capture temporal dynamics.
- Dense and complex figure panels. Simplify or separate figures into sub-panels, use arrows or labels to highlight key findings, and avoid overcrowding.
- No visual summary of findings. Add a graphical abstract or summary diagram illustrating the main findings and proposed mechanism of T14 regulation.
- Lack of detail on normalization and blinding. Clarify whether image analysis was blinded and specify normalization procedures for intensity measurements.
- Manual thresholding may introduce bias. Use automated thresholding algorithms and report the parameters used to ensure reproducibility.
- No assessment of T14's physiological effects. Include in vitro or in vivo assays to test whether altered T14 levels affect neuron health, signaling, or connectivity.
- Limited discussion on translational implications. Expand the discussion to compare findings with human Alzheimer’s pathology and explore clinical relevance.
Author Response
We thank the referees for their constructive criticisms on our manuscript entitled "Widespread Changes of Bioactive Peptide T14 Immunoreactivity after Manipulation of Cortical Projection Neuron Activity".
We were pleased to read your overall comments “the study addresses an important and timely topic in the field of neurobiology and Alzheimer’s disease, particularly by exploring the activity-dependent regulation of T14, a peptide with emerging significance. The methodological design is conceptually sound, and the results are of clear potential interest to the field.”
We thank you and the referees for the suggestions for revisions to improve the rigor, clarity, and translational relevance of the study. Since we had 10 days to perform these revisions, we could not entertain some of the suggested experimental ideas, some would take years of work, but we shall follow these suggestions up in the future.
Reviewer 1
Comment 1: Small sample sizes (n=2–3 mice per group). Increase the number of animals per group to at least 5–6 and perform a power analysis to justify sample sizes.
Answer 1: We thank the reviewer for raising this important point regarding sample size. We fully agree that larger group sizes (n = 5–6) would strengthen the statistical power of the study. However, the current experiments were conducted on the same cohort of animals used in our previously published work (Vadisiute et al., Commun Biol 2024; https://doi.org/10.1038/s42003-024-06994-w), which included extensive physiological and anatomical validations. As such, the current findings are supported by a robust foundation of prior data. As detailed in our previous publication, the number of animals was limited due to welfare concerns: higher doses of CNO led to adverse effects in the animals, and we made a deliberate ethical decision to minimize further use of this cohort. Additionally, the transgenic mouse line used in these experiments is no longer being bred, and we are not in a position to re-establish the colony at this time. We acknowledge this as a limitation of the current study and have added a note in the revised manuscript to reflect this.
Comment 2: Lack of biological replicates across batches. Repeat key experiments in at least two independent experimental batches or litters to ensure reproducibility.
Answer 2: Please find attached information about animals used for this study. We provided detailed information about animal genotype, sex and litters. This information can be found in supplementary material file “Source data” → “Animal information”

Comments 3: No validation of the anti-T14 antibody specificity. Perform pre-adsorption controls, Western blots using brain tissue, or test the antibody on T14 knockout samples.
Answer 3: Neuro-Bio has previously shown the specificity and validation of the T14 antibody. In Figure 1 from the paper Rocha et al, 2023, the authors show how the antibody only recognises T14 and the epitope is VHWK specifically, see figure below. Also, The supplementary materials for our 2022 paper specifically cover pre-adsorption. We wrote: "For the immunoneutralization experiments, 1 µg/ml of T14 in 1:200 dilution of anti-T14 was used." In the Results section of the papers we wrote: "The immunoreactive signal for T14 was lost following neutralization of the primary antibody with the peptide (Figure 1G, I, K)."

Specificity of T14 antibody. (a) Comparison of T14 antibody specificity using ELISA to detect peptides of various lengths with the C-terminus capped with W or K. (b) List of peptides used with amino acids sequences (epitope of the antibody -VHWK highlighted in red). (c) Dose-response of different peptides at the nanomolar range to determine the importance of the length and the epitope. (d) Dose response of T14, T30, T15 and AChE at nanomolar range. (d) Effect of trypsin on full AChE, cleaving T14 now detectable by the antibody.
In the same paper, the authors have shown that by immunoneutralisation, the signal of the T14 antibody dissappears.

Peptide block of T14 and anti-T14 staining of young PP skin sample and aged PP skin sample. Peptide successfully blocked T14 binding of epitope for both young and aged photo-protected skin tissues.
Comment 4: No biochemical confirmation of T14 distribution. Include Western blot or ELISA analyses to confirm the presence and size of T14 and support immunostaining results.
Answer 4: The authors have published in several occasions the detection of T14 in different areas of the brain using WB and ELISA, for example in the following articles: Garcia-Ratés et al 2016, Greenfield, Cole, Coen et al 2022, Greenfield, Ferrari, Coen et al. 2022.
Comment 5: Indirect assessment of intracellular vs. extracellular localization. Use immunogold electron microscopy or subcellular fractionation to directly visualize the location of T14.
Answer 5: We appreciate Reviewer 1’s suggestion regarding the direct assessment of T14 localization. We agree that techniques such as immunogold electron microscopy or subcellular fractionation would provide more definitive insights into the intracellular versus extracellular distribution of T14. While these methods were beyond the scope of the current study, we acknowledge this as a limitation and will incorporate these approaches in future investigations to more precisely determine the localization of T14.
Comments 6: Co-localization analysis is largely descriptive. Quantify co-localization using Manders’ or Pearson’s coefficients and include appropriate statistical controls.
Answer 6: In our study, we employed an object-based 3D co-localisation analysis, which is particularly well-suited for high-resolution 3D confocal datasets. Unlike pixel-based methods such as Pearson’s or Manders’ coefficients—which assess intensity correlations across all image pixels—object-based methods identify discrete structures (e.g., puncta or cell compartments) in each channel and quantify their spatial relationships. This approach is widely regarded as more accurate for 3D data, as it avoids confounding effects from background noise and out-of-focus light.
While we acknowledge the utility of Pearson’s and Manders’ coefficients for certain applications, we chose object-based analysis because it better reflects biologically meaningful co-localisation in our 3D images. We have clarified this rationale in the revised manuscript and added a brief explanation of the method used in the Methods section to improve transparency.
Comment 7: No discussion of compensatory mechanisms in Snap25 Cko. Add a paragraph discussing potential compensatory changes due to early developmental silencing and suggest using inducible models in the future.
Answer 7: We appreciate the reviewer’s suggestion to include this into discussion.
line 375-381: It appears that chronic silencing exerts a stronger effect on T14 than acute manipulations of neuronal activity. This suggests that prolonged disruption of synaptic function leads to more pronounced alterations in T14 expression or localization. In the absence of SNAP25, other SNARE proteins or synaptic components may partially compensate for the loss; however, this compensation is often incomplete and insufficient to fully restore normal synaptic function, which may contribute to the progressive neuronal dysfunction observed.
Comment 8: No developmental time-course for T14 alterations. Include intermediate timepoints (e.g., P14, P30) to monitor the dynamics of T14 expression across development.
Answer 8: We appreciate the reviewer’s insightful suggestion regarding the inclusion of intermediate developmental timepoints to better characterize the dynamics of T14 expression. We agree that a full developmental time-course, including additional stages such as P14 and P30, would provide a more comprehensive understanding of how T14 levels evolve over time.In the current study, our focus was to compare T14 expression at key developmental stages representing embryonic and early postnatal (P7) development and mature (P60–P90) timepoints. These ages were chosen based on prior work demonstrating critical transitions in cortical development and circuit maturation. While this approach captured meaningful differences between immature and mature states, we recognize that intermediate timepoints could help delineate the trajectory and rate of T14 expression changes more precisely.
Comment 9: No functional or behavioral readouts. Incorporate behavioral tests (e.g., open field, rotarod) or physiological recordings to connect molecular changes with brain function.
Answer 9: Functional and behaviour analysis of Rbp4-Cre;Snap25 cKO mice was already described in previous publications: Washbourne, P., Thompson, P., Carta, M. et al. Genetic ablation of the t-SNARE SNAP-25 distinguishes mechanisms of neuroexocytosis. Nat Neurosci 5, 19–26 (2002). https://doi.org/10.1038/nn783; Krone, L.B., Yamagata, T., Blanco-Duque, C. et al. A role for the cortex in sleep–wake regulation. Nat Neurosci 24, 1210–1215 (2021). https://doi.org/10.1038/s41593-021-00894-6. We also have new behaviour data analysis but publication is in preparation.
Comment 10: No mechanistic explanation for changes in the substantia nigra. Discuss possible indirect circuit mechanisms (e.g., polysynaptic pathways or diffusible factors) and consider using retrograde tracing.
Answer 10: We thank the reviewer for raising this important point. As noted in the manuscript, there is no known direct projection from the manipulated cortical region to the substantia nigra (SN) (Rbp4 subpopulation of layer 5 projection neurons does not have projections to substantia nigra), which makes the observed changes in the SN particularly intriguing. We agree that indirect mechanisms, such as modulation via polysynaptic pathways, network-level plasticity, or the action of diffusible signaling molecules, are plausible explanations. Although these mechanisms remain speculative in the current context, they warrant further investigation. We also appreciate the suggestion to consider retrograde tracing approaches to map potential indirect circuit pathways more precisely. This is an excellent direction for future studies and will be essential to better understand the underlying connectivity and communication between these regions. At this stage, we believe the inclusion of the SN findings provides valuable preliminary insight into the broader network-level effects of our manipulation, even if the mechanistic basis remains unclear.
Comment 11: No downstream pathway analysis. Investigate the impact of T14 on calcium signaling, α7 receptor activation, or downstream gene expression using pharmacological tools.
Answer 11: We appreciate the suggestions to investigate T14 on calcium signalling, alfa7 receptor activation and gene expression using pharmacological tools. This is an excellent direction for future studies and better understanding T14 role as potential signalling molecule.
Comment 12: Single-dose/single-timepoint chemogenetic manipulations. Explore a dose–response curve and test multiple timepoints (e.g., 30 min, 2h, 6h) to capture temporal dynamics.
Answer 12: We thank the reviewer for this thoughtful comment. As noted, the current study used a single-dose, single-timepoint chemogenetic manipulation protocol, consistent with our previously published work (Vadisiute et al., Commun Biol 2024; https://doi.org/10.1038/s42003-024-06994-w). In that study, we observed rapid and robust glial and neuronal responses following acute DREADD activation, and the 90-minute post-injection timepoint was selected based on both the peak of these responses and ethical considerations regarding animal wellbeing. Higher doses or longer durations of CNO exposure were associated with adverse effects, which limited our ability to explore a broader dose–response range within the constraints of the transgenic mouse line used.
We agree that establishing a dose–response curve and assessing multiple timepoints (e.g., 30 min, 2 h, 6 h) would provide valuable insight into the temporal dynamics of the observed effects. We are currently exploring the use of viral DREADD delivery, which offers greater experimental flexibility, including the potential for longitudinal assessments at multiple timepoints and with varying levels of DREADD expression and activation.
Comment 13: Dense and complex figure panels. Simplify or separate figures into sub-panels, use arrows or labels to highlight key findings, and avoid overcrowding.
Answer 13: We appreciate the reviewer’s suggestion regarding the figure panels. After careful consideration, we believe the current layout is necessary to preserve the continuity and comprehensive nature of the data. However, to improve clarity, we prepared a graphic abstract highlighting our main findings and summarising results. We hope this addresses the concern while maintaining the integrity of the data presentation.
Comment 14: No visual summary of findings. Add a graphical abstract or summary diagram illustrating the main findings and proposed mechanism of T14 regulation.
Answer 14: we included graphical abstract

Comment 15: Lack of detail on normalization and blinding. Clarify whether image analysis was blinded and specify normalization procedures for intensity measurements.
Answer 15: We thank the reviewer for pointing out the need for more detail regarding blinding and normalization procedures. For image analysis, we acknowledge that full blinding was not feasible in all experiments due to the pronounced nature of the observed phenotypes, which often made group identity evident during analysis. However, wherever possible, automated or semi-automated analysis tools were used to minimize potential bias. Regarding normalization, intensity measurements were normalised to control signal levels. This approach was applied consistently across all experimental groups and imaging sessions.
Comment 16: Manual thresholding may introduce bias. Use automated thresholding algorithms and report the parameters used to ensure reproducibility.
Answer 16: In our analysis, manual thresholding was used to detect T14 immunoreactivity, particularly because in older animals T14 tends to form distinct clusters, making automated thresholding less reliable in differentiating signal from background in these cases. That said, we recognise the potential for bias introduced by manual thresholding and agree that automated methods would improve reproducibility.
Comment 17: No assessment of T14's physiological effects. Include in vitro or in vivo assays to test whether altered T14 levels affect neuron health, signaling, or connectivity.
Answer 17: We thank the reviewer for this excellent suggestion. We agree that assessing the physiological effects of T14 levels would enhance our understanding of T14's functional role. While these experiments were beyond the scope of the current study, which focused primarily on T14 expression and localization, we recognize the importance of linking molecular changes to functional outcomes. In future work, we plan to incorporate in vitro and in vivo electrophysiological recordings to directly assess how modulation of T14 levels influences neuronal activity and network connectivity.
Comment 18: Limited discussion on translational implications. Expand the discussion to compare findings with human Alzheimer’s pathology and explore clinical relevance.
Answer 18: As noted in the manuscript, we have previously demonstrated that T14 levels vary across Braak stages, correlating with Alzheimer’s pathology progression (Greenfield, Cole, Coen et al., 2022). Additionally, our recent review (Garcia-Ratés et al., 2024) thoroughly explores the potential of T14 as a biomarker for Alzheimer’s disease, highlighting its translational relevance. We also have just resubmitted a paper to the NEJM after comments from reviewers on the use of the peptide in saliva to diagnose the disease.
Reviewer 2 Report
Comments and Suggestions for Authors
In the manuscript (ID: ijms-3662257), the authors studied the extensive changes in the immune reactivity of bioactive peptide T14 after manipulating the activity of cortical projection neurons. In general, the research meets the requirements of International Journal of Molecular Sciences. However, the overall quality of the manuscript is concerning, and it is recommended to revise it significantly before publication in International Journal of Molecular Sciences. Below are some suggestions and issues for the authors to consider and improve.
(1) Title: The meaning of the title is not very clear. It is suggested that the authors make some revisions. Should be “Widespread Changes in the Immunoreactivity of Bioactive Peptide T14 after Manipulating the Activity of Cortical Projection Neuron” rather “Widespread Changes of Bioactive Peptide T14 Immunoreactivity after Manipulation of Cortical Projection Neuron Activity”.
(2) Abstract: The abstract should focus on introducing the results and conclusions of this study. Therefore, it is suggested that the author reasonably reduce the content of the experimental methods in the abstract and introduce the results in more detail.
(3) Keywords: These words including T14, Snap25, Rbp4-Cre, and DREADD are not suitable as keywords. It is suggested that the author provide their full names, which will be more conducive to readers' understanding of this manuscript.
(4) Keywords: It is suggested that the authors add this keyword of “Immunoreactivity”.
(5) 1. Introduction: In this part, the author lacks a relatively systematic review of the background and innovation of this research, which leads to the relatively poor readability of this manuscript. It is suggested that the author carefully summarize the related research and further explain the innovation and significance of this study.
(6) Line 35 and 51: “in vitro” and “in vivo” should be italic and please revise the mistake. In addition, there are similar errors in other parts of the manuscript, and authors are advised to check the whole manuscript carefully and correct these minor errors.
(7) Line 35: entry1,2. The citation style of references in the manuscript does not comply with the rules of International Journal of Molecular Sciences. Please make thorough revisions to them in accordance with the requirements of the journal.
(8) Line 37: Should be “acts as” rather “functions as”.
(9) Line 80-88: To examine the distribution of T14 during normal development … or without Triton (-Triton), where the membrane remains intact. It is suggested that the author directly state the specific experimental results in the results section instead of elaborating on the experimental methods at length.
(10) 3. Discussion: The author has conducted a relatively detailed discussion on the obtained research results and provided explanations for future research
(11) 4. Methods: Each method written in the manuscript should include the reference used, which allows the reader to access the original experimental methods.
(12) This manuscript has omitted the Conclusion. It is suggested that the authors supplement it.
Author Response
We thank the referees for their constructive criticisms on our manuscript entitled "Widespread Changes of Bioactive Peptide T14 Immunoreactivity after Manipulation of Cortical Projection Neuron Activity".
We were pleased to read your overall comments “the study addresses an important and timely topic in the field of neurobiology and Alzheimer’s disease, particularly by exploring the activity-dependent regulation of T14, a peptide with emerging significance. The methodological design is conceptually sound, and the results are of clear potential interest to the field.”
We thank you the referees for the suggestions for revisions to improve the rigor, clarity, and translational relevance of the study. Since we had 10 days to perform these revisions, we could not entertain some of the suggested experimental ideas, some would take years of work, but we shall follow these suggestions up in the future.
Comments (1): Title: The meaning of the title is not very clear. It is suggested that the authors make some revisions. Should be “Widespread Changes in the Immunoreactivity of Bioactive Peptide T14 after Manipulating the Activity of Cortical Projection Neuron” rather “Widespread Changes of Bioactive Peptide T14 Immunoreactivity after Manipulation of Cortical Projection Neuron Activity”.
Answer 1: We would like to thank reviewer for this observation and we changed our title to Lines 1-2: “Widespread Changes in the Immunoreactivity of Bioactive Peptide T14 after Manipulating the Activity of Cortical Projection Neuron”
Comment (2): Abstract: The abstract should focus on introducing the results and conclusions of this study. Therefore, it is suggested that the author reasonably reduce the content of the experimental methods in the abstract and introduce the results in more detail.
Answer 2: We agree with reviewer and we changed our abstract accordingly:
Lines 21-38: Previous studies have suggested that T14, a 14-amino acid peptide derived from acetylcholinesterase (AChE), functions as an activity-dependent signalling molecule with key roles in brain development, and its dysregulation has been linked to neurodegeneration in Alzheimer’s disease. In this study, we examined the distribution of T14 under normal developmental conditions in the mouse forebrain, motor cortex (M1), striatum (STR), and substantia nigra (SN). T14 immunoreactivity declined from E16 to E17 and further decreased by P0, then peaked at P7 during early postnatal development before declining again by adulthood at P70. Lower T14 immunoreactivity in samples processed without Triton indicated that T14 is primarily localized intracellularly. To explore the relationship between T14 expression and neuronal activity, we used mouse models with chronic silencing (Rbp4Cre-Snap25), acute silencing (Rbp4Cre-hM4Di), and acute activation (Rbp4Cre-hM3D1). Chronic silencing altered the location and size of intracellular T14-immunoreactive particles in adult brains, while acute silencing had no observable effect. In contrast, acute activation increased T14+ density in the STR, modified T14 puncta size near Rbp4Cre cell bodies in M1 layer 5 and their projections to the STR, and enhanced co-localization of T14 with presynaptic terminals in the SN.
Comment (3)/(4):
(3) Keywords: These words including T14, Snap25, Rbp4-Cre, and DREADD are not suitable as keywords. It is suggested that the author provide their full names, which will be more conducive to readers' understanding of this manuscript.
(4) Keywords: It is suggested that the authors add this keyword of “Immunoreactivity”.
Answer 3/4: we provided full names and included “immunoreactivity”, lines 40-43: 14-mer peptide (T14), Synaptosomal associated protein 25 kDa (Snap25), retinol binding protein 4 (Rbp4), Designer Receptors Exclusively Activated by Designer Drugs (DREADDs), developing brain, Alzheimer’s Disease, immunoreactivity
Comment (5): 1. Introduction: In this part, the author lacks a relatively systematic review of the background and innovation of this research, which leads to the relatively poor readability of this manuscript. It is suggested that the author carefully summarize the related research and further explain the innovation and significance of this study.
Answer 5: We have revised the Introduction section to provide a more systematic and comprehensive review of the relevant background literature. Additionally, we have clearly highlighted the innovation and significance of our study to improve the overall readability and contextual understanding of the manuscript. We believe these changes address your concerns and enhance the clarity and impact of the Introduction.
Lines 46-59: T14 is a peptide derived from the widespread enzyme acetylcholinesterase (AChE) , but independently bioactive of the parent molecule and cholinergic transmission (Day and Greenfield, 2002). T14 promotes cell growth via the mTOR pathway, and declines with normal ageing (Garcia-Ratés et al 2024). It acts exclusively at an allosteric site on alpha-7 nicotinic receptors to increase Ca²⁺ entry in the presence of the primary ligand acetylcholine (ACh) (Ranglani et al., 2024; Garcia-Ratés et al., 2024). Its bioactivity has been demonstrated in vitro and ex vivo using PC12 cells and rodent brain slices. It has been hypothesized that T14 acts as an activity-dependent signalling molecule, promoting cell growth and renewal during brain development but when inappropriately activated in the mature brain, it may instead be the key molecule driving the process of neurodegeneration in diseases such as Alzheimer’s disease (AD) (Garcia-Ratés et al., 2024). In this context, previous in vivo studies in the 5XFAD mouse model of AD, have shown elevated T14 immunoreactivity in the rodents brains in the areas implicated in neurodegenerative diseases (Greenfield et al., 2022a; Greenfield et al., 2022b).
Comment (6): Line 35 and 51: “in vitro” and “in vivo” should be italic and please revise the mistake. In addition, there are similar errors in other parts of the manuscript, and authors are advised to check the whole manuscript carefully and correct these minor errors.
Answer 6: We have carefully reviewed the entire manuscript and corrected all instances of “in vitro” and “in vivo” by formatting them in italics as in vitro and in vivo. We also checked for similar errors throughout the manuscript to ensure consistency and accuracy.
Comment (7): Line 35: entry1,2. The citation style of references in the manuscript does not comply with the rules of International Journal of Molecular Sciences. Please make thorough revisions to them in accordance with the requirements of the journal.
Answer 7: We have carefully revised the citation style of all references in the manuscript to fully comply with the guidelines of the International Journal of Molecular Sciences.
Comment (8): Line 37: Should be “acts as” rather “functions as”.
Answer 8: We have revised the text by changing “functions as” to “acts as” as suggested.
Comment (9): Line 80-88: To examine the distribution of T14 during normal development … or without Triton (-Triton), where the membrane remains intact. It is suggested that the author directly state the specific experimental results in the results section instead of elaborating on the experimental methods at length.
Answer 9: We explained experimental methods in more detail. Lines 106-117: To investigate the cellular localisation of T14, we performed immunohistochemistry on free-floating coronal brain sections under two different conditions to distinguish between intracellular and extracellular localization. Sections were incubated in blocking solutions containing either 10% normal goat serum (NGS) alone or 10% NGS with 0.3% Triton X-100, a detergent commonly used to permeabilize cell membranes. The presence of Triton (+Triton) disrupts the integrity of the plasma membrane, allowing antibodies to access intracellular compartments. This condition enables the detection of both intracellular and extracellular antigens. In contrast, the absence of Triton (-Triton) preserves the membrane barrier, limiting antibody access to extracellular or membrane-associated antigens only. By comparing T14 immunoreactivity between these two conditions, we can infer its subcellular localisation.
Comment (10): 3. Discussion: The author has conducted a relatively detailed discussion on the obtained research results and provided explanations for future research
Comment (11): 4. Methods: Each method written in the manuscript should include the reference used, which allows the reader to access the original experimental methods.
Comment (12) This manuscript has omitted the Conclusion. It is suggested that the authors supplement it.
Answer 12: We included conclusions to our manuscript. Lines 382-399:
We investigated the bioactive peptide T14, which has both trophic and toxic roles depending on developmental or pathological context. T14 immunoreactivity peaks at postnatal day 7 and varies across developmental stages. Chronic silencing of cortical layer 5 neurons in the Snap25 cKO model caused region-specific changes in T14 localization, density, and granule size, especially in motor cortex and striatum, while acute silencing had little effect. Acute activation increased T14 levels and altered its association with presynaptic terminals.
Ultrastructural analysis revealed lysosome-like structures in degenerating neurites that may contain T14, linking chronic synaptic disruption to neuronal degeneration. These changes were not seen after acute manipulations. Partial compensation by other SNARE proteins occurs in the absence of SNAP25 but is insufficient to prevent progressive dysfunction.
Our study underscores the complex role of T14 in neuronal development and pathology, highlighting the importance of synaptic activity patterns in modulating its function. Future work focusing on the precise subcellular localization of T14 granules and the mechanisms underlying neuronal death in Snap25-deficient neurons will provide deeper insight into its dual roles and therapeutic potential.
Round 2
Reviewer 2 Report
Comments and Suggestions for Authors
The quality of the manuscript (ijms-3662257) has been improved accordingly, and I think that the manuscript can be accepted for publication in International Journal of Molecular Sciences.